# A Jacobi–Davidson Method for Large Scale Canonical Correlation Analysis

**Zhongming Teng *** **and Xiaowei Zhang**

College of Computer and Information Science, Fujian Agriculture and Forestry University,
Fuzhou 350002, China; xwzhang.fafu@gmail.com
* Correspondence: zhmteng@fafu.edu.cn

**Abstract:** In the large scale canonical correlation analysis arising from multi-view learning applications, one needs to compute canonical weight vectors corresponding to a few of largest canonical correlations. For such a task, we propose a Jacobi–Davidson type algorithm to calculate canonical weight vectors by transforming it into the so-called canonical correlation generalized eigenvalue problem. Convergence results are established and reveal the accuracy of the approximate canonical weight vectors. Numerical examples are presented to support the effectiveness of the proposed method.

**Keywords:** canonical correlation analysis; Jacobi–Davidson; generalized eigenvalue problems; convergence

## 1. Introduction

Canonical correlation analysis (CCA) is one of the most representative two-view multivariate statistical techniques and has been applied to a wide range of machine learning problems including classification, retrieval, regression and clustering [1–3]. It seeks a pair of linear transformations for two view high-dimensional features such that the associated low-dimensional projections are maximally correlated. Denote the data matrices $S_a \in \mathbb{R}^{m \times d}$ and $S_b \in \mathbb{R}^{n \times d}$ from two data sets with $m$ and $n$ features, respectively, where $d$ is the number of samples. Without loss of generality, we assume $S_a$ and $S_b$ are centered, i.e., $S_a \mathbf{1}_d = 0$ and $S_b \mathbf{1}_d = 0$ where $\mathbf{1}_d \in \mathbb{R}^d$ is the vector of all ones, otherwise, we can preprocess $S_a$ and $S_b$ as $S_a \leftarrow S_a - \frac{1}{d}(S_a \mathbf{1}_d)\mathbf{1}_d^{\mathrm{T}}$ and $S_b \leftarrow S_b - \frac{1}{d}(S_b \mathbf{1}_d)\mathbf{1}_d^{\mathrm{T}}$, respectively. CCA aims to find a pair of canonical weight vectors $x \in \mathbb{R}^m$ and $y \in \mathbb{R}^n$ that maximize the canonical correlation

$$\max x^{\mathrm{T}} C y, \quad \text{subject to} \quad x^{\mathrm{T}} A x = 1 \text{ and } y^{\mathrm{T}} B y = 1, \tag{1}$$

where

$$A = S_a S_a^{\mathrm{T}} \in \mathbb{R}^{m \times m}, \quad B = S_b S_b^{\mathrm{T}} \in \mathbb{R}^{n \times n} \quad \text{and} \quad C = S_a S_b^{\mathrm{T}} \in \mathbb{R}^{m \times n}, \tag{2}$$

and then projects the high-dimensional data $S_a$ and $S_b$ onto low-dimensional subspaces spanned by $x$ and $y$, respectively, to achieve the purpose of dimensionality reduction. In most cases [1,4,5], only one pair of canonical weight vectors is not enough since it means the dimension of low-dimensional subspaces is just one. When a set of canonical weight vectors are required, the single-vector CCA (1) has been extended to obtain the pair of canonical weight matrices $X \in \mathbb{R}^{m \times k}$ and $Y \in \mathbb{R}^{n \times k}$ by solving the optimization problem

$$\max \operatorname{trace}(X^{\mathrm{T}} C Y), \quad \text{subject to} \quad X^{\mathrm{T}} A X = I_k \text{ and } Y^{\mathrm{T}} B Y = I_k. \tag{3}$$

Usually, both $A$ and $B$ are symmetric positive definite. However, there are cases, such as the under-sampled problem [6], that $A$ and $B$ may be semi-definite. In such a case, some regular techniques [7–10] by adding a multiple of the identity matrix to them are applied to find the optimal solution of

$$\max \ \text{trace}(X^{\mathrm{T}}CY), \quad \text{subject to} \quad X^{\mathrm{T}}(A + \kappa_a I_m)X = I_k \text{ and } Y^{\mathrm{T}}(B + \kappa_b I_n)Y = I_k,$$

where $\kappa_a$ and $\kappa_b$ are called regularization parameters and they usually are chosen to maximize the cross-validation score [11]. In other words, $A$ and $B$ are replaced by $A + \kappa_a I_m$ and $B + \kappa_b I_n$ to keep the invertible of $A$ and $B$, respectively. Hence, in this paper, by default we assume $A$ and $B$ are both positive definite and $m \geq n$ unless explicitly stated otherwise.

As shown in [4], the optimization problem (3) can be equivalently transformed into solving the following generalized eigenvalue problem

$$Kz := \begin{bmatrix} 0 & C \\ C^{\mathrm{T}} & 0 \end{bmatrix} \begin{bmatrix} x \\ y \end{bmatrix} = \lambda \begin{bmatrix} A & 0 \\ 0 & B \end{bmatrix} \begin{bmatrix} x \\ y \end{bmatrix} = \lambda Mz, \tag{4}$$

where the positive definiteness of the matrices $A$ and $B$ implies $M$ being positive definite. The generalized eigenvalue problem (4) is referred as the Canonical Correlation Generalized Eigenvalue Problem (CCGEP) in this paper. Define $J := \begin{bmatrix} R_{\mathrm{A}}^{-1} & 0 \\ 0 & R_{\mathrm{B}}^{-1} \end{bmatrix}$ where

$$A = R_{\mathrm{A}}^{\mathrm{T}}R_{\mathrm{A}} \quad \text{and} \quad B = R_{\mathrm{B}}^{\mathrm{T}}R_{\mathrm{B}} \tag{5}$$

are their Cholesky decomposition. It is easy to verify that

$$J^{\mathrm{T}} \begin{bmatrix} 0 & C \\ C^{\mathrm{T}} & 0 \end{bmatrix} J J^{-1} \begin{bmatrix} x \\ y \end{bmatrix} = \lambda J^{\mathrm{T}} \begin{bmatrix} A & 0 \\ 0 & B \end{bmatrix} J J^{-1} \begin{bmatrix} x \\ y \end{bmatrix}$$

gives rise to

$$\begin{bmatrix} 0 & \widetilde{C} \\ \widetilde{C}^{\mathrm{T}} & 0 \end{bmatrix} \begin{bmatrix} p \\ q \end{bmatrix} = \lambda \begin{bmatrix} p \\ q \end{bmatrix}, \tag{6}$$

where

$$\widetilde{C} = R_{\mathrm{A}}^{-\mathrm{T}}CR_{\mathrm{B}}^{-1}, \quad p = R_{\mathrm{A}}x \quad \text{and} \quad q = R_{\mathrm{B}}y, \tag{7}$$

and it implies

$$\widetilde{C}q = \lambda p \quad \text{and} \quad \widetilde{C}^{\mathrm{T}}p = \lambda q.$$

It means that the eigenpairs of (4) can be obtained by computing the singular values and the associated left and right singular vectors of $\widetilde{C}$. This method works well when the sample size $d$ and feature dimension $m$ and $n$ are of moderate size but it will be very slow and numerically unstable for large-scale datasets which are ubiquitous in the age of "Big Data" [12]. For large-scale datasets, the equivalence between (4) and (6) makes it possible for us to simply adapt the subspace type algorithms for calculating the partial singular values decomposition, such as Lanczos type algorithms [13,14] and Davidson type algorithms [15,16], and then translate them for CCGEP (4). However, in practice, the decompositions of the sample covariance matrices $A$ and $B$ are usually unavailable in large scale matrix cases. The reason is that the decomposition is too expensive to compute explicitly for large scale problems, and may destroy the sparsity and some structural information. Furthermore, sometime sample covariance matrices $A$ and $B$ should never be explicitly formed, such as in online learning systems.

Meanwhile, in [17], it is suggested to solve CCGEP (4) by considering the large scale symmetric positive definite pencil $\{K, M\}$. Some subspace type numerical algorithms also have been generalized to computing partial eigenpairs of $\{K, M\}$, see [18,19]. However, these generic algorithms do not make use of the special structure in (4), and they usually are less efficient than custom-made algorithms. Therefore, existing algorithms either can not avoid the covariance matrices decomposition, or do not consider the structure of CCGEP.

In this paper, we will focus on the Jacobi–Davidson type subspace method for canonical correlation analysis. The idea of Jacobi–Davidson algorithm proposed in [20] is Jacobi's approach combined with Davidson type subspace method. Its essence is to construct a correction for a given eigenvector approximation in a subspace orthogonal to the given approximation. The correction in a given subspace is extracted in a Davidson manner, and then the expansion of the subspace is done by solving its correction equation. Due to the significant improvement in convergence, the Jacobi–Davidson has been one of the most powerful algorithms in the matrix eigenvalue problem, and is almost generalized to all fields of matrix computation. For example, in [15,21], Hochstenbach presented Jacobi–Davidson methods for singular value problems and generalized singular value problems, respectively. In [22,23], Jacobi–Davidson methods are developed to solve the nonlinear and two-parameter eigenvalue problems, respectively. Some other recent work on Jacobi–Davidson methods can be found in [24–29]. Motivated by these facts, we will continue the effort by extending the Jacobi–Davidson variant to canonical correlation analysis. The main contribution is that the algorithm directly tackles CCGEP (4) without involving the large matrix decomposition, and does take advantage of the special structure of $K$ and $M$, while the significance of transforming (4) into (6) lies only in our theoretical developments.

The rest of this paper is organized as follows. Section 2 collects some notations and a basic result for CCGEP that are essential to our later development. Our main algorithm is given and analyzed in detail in Section 3. We present some numerical examples in Section 4 to show the behaviors of our proposed algorithm and to support our analysis. Finally, conclusions are made in Section 5.

## 2. Preliminaries

Throughout this paper, $\mathbb{R}^{m \times n}$ is the set of all $m \times n$ real matrices, $\mathbb{R}^m = \mathbb{R}^{m \times 1}$, and $\mathbb{R} = \mathbb{R}^1$. $I_n$ is the $n \times n$ identity matrix. The superscript "$\cdot^{\mathrm{T}}$" takes transpose only, and $\| \cdot \|_1$ denotes the $\ell_1$-norm of a vector or matrix. For any matrix $N \in \mathbb{R}^{m \times n}$ with $m \geq n$, $\sigma_i(N)$ for $i = 1, \ldots, n$ is used to denote the singular values of $N$ in descending order.

For vectors $x, y \in \mathbb{R}^n$, the usual inner product and its induced norm are conveniently defined by

$$\langle x, y \rangle := y^{\mathrm{T}} x, \quad \|x\|_2 := \sqrt{\langle x, x \rangle}.$$

With them, the usual acute angle $\angle(x, y)$ between $x$ and $y$ can then be defined by

$$\angle(x, y) := \arccos \frac{|\langle x, y \rangle|}{\|x\|_2 \|y\|_2}.$$

Similarly, given any symmetric positive definite $W \in \mathbb{R}^{n \times n}$, the $W$-inner product and its induced $W$-norm are defined by

$$\langle x, y \rangle_w := y^{\mathrm{T}} W x, \quad \|x\|_w := \sqrt{\langle x, x \rangle}_w.$$

If $\langle x, y \rangle_w = 0$, then we say $x \perp_w y$ or $y \perp_w x$. The $W$-acute angle $\angle_w(x, y)$ between $x$ and $y$ can then be defined by

$$\angle_w(x, y) := \arccos \frac{|\langle x, y \rangle_w|}{\|x\|_w \|y\|_w}.$$

Let the singular value decomposition of $\widetilde{C}$ be $\widetilde{C} = P \Lambda Q^{\mathrm{T}}$ where $P = [p_1, \ldots, p_m] \in \mathbb{R}^{m \times m}$ and $Q = [q_1, \ldots, q_n] \in \mathbb{R}^{n \times n}$ are orthonormal, i.e., $P^{\mathrm{T}} P = I_m$ and $Q^{\mathrm{T}} Q = I_n$, and $\Lambda = \mathrm{diag}(\lambda_1, \ldots, \lambda_n) \in \mathbb{R}^{m \times n}$ with $\lambda_1 \geq \lambda_2 \geq \cdots \geq \lambda_n \geq 0$ is a leading diagonal matrix. The singular value decomposition of $\widetilde{C}$ closely relates to the eigendecomposition of the following symmetric matrix [30] (p. 32):

$$\begin{bmatrix} 0 & \widetilde{C} \\ \widetilde{C}^{\mathrm{T}} & 0 \end{bmatrix}, \tag{8}$$

whose eigenvalues are $\pm\lambda_i$ for $i = 1, \ldots, n$ plus $m - n$ zeros, i.e.,

$$-\lambda_1 \le \cdots \le -\lambda_n \le \underbrace{0 \le \cdots \le 0}_{m-n} \le \lambda_n \le \cdots \le \lambda_1, \tag{9}$$

with associated eigenvectors are

$$\begin{bmatrix} p_i \\ \pm q_i \end{bmatrix}, \ i = 1, 2, \ldots, n, \quad \text{and} \quad \begin{bmatrix} p_i \\ 0 \end{bmatrix}, \ i = n+1, \ldots, m,$$

respectively. The equivalence between (4) and (6) leads that the eigenvalues of CCGEP (4) are $\pm\lambda_i$ for $i = 1, \ldots, n$ plus $m - n$ zeros, and the corresponding eigenvectors are

$$\begin{bmatrix} x_i \\ \pm y_i \end{bmatrix}, \ i = 1, 2, \ldots, n, \quad \text{and} \quad \begin{bmatrix} x_i \\ 0 \end{bmatrix}, \ i = n+1, \ldots, m,$$

respectively, where

$$x_i = R_{\text{A}}^{-1} p_i \quad \text{and} \quad y_i = R_{\text{B}}^{-1} q_i. \tag{10}$$

Let $X = [x_1, \ldots, x_m]$ and $Y = [y_1, \ldots, y_n]$. Then, the $A$- and $B$-orthonormal constraints of $X$ and $Y$, respectively, i.e.,

$$X^{\text{T}} A X = I_m \quad \text{and} \quad Y^{\text{T}} B Y = I_n \tag{11}$$

are followed by $P^{\text{T}} P = I_m$ and $Q^{\text{T}} Q = I_n$. Here, the first few $x_i$ and $y_i$ for $i = 1, 2, \ldots, k$ with $k < n$ are wanted canonical correlation weight vectors. Furthermore, their corresponding eigenvalues satisfy the following maximization principle which is critical to our later developments. For the proof see Appendix A.1.

**Theorem 1.** *The following equality holds for any $U \in \mathbb{R}^{m \times k}$ and $V \in \mathbb{R}^{n \times \ell}$*

$$\sum_{i=1}^{\min\{k,\ell\}} \lambda_i = \max_{U^{\text{T}} A U = I_k, V^{\text{T}} B V = I_\ell} \sum_{i=1}^{\min\{k,\ell\}} \sigma_i \left( U^{\text{T}} C V \right), \tag{12}$$

*where $A$, $B$ and $C$ are defined in (2) and $\lambda_i$ defined in (9), and $\sigma_i(U^{\text{T}} C V)$ for $1 \le i \le \min\{k, \ell\}$ are the singular values of $U^{\text{T}} C V$.*

## 3. The Main Algorithm

The idea of the Jacobi–Davidson method [20] is to construct iteratively approximations of certain eigenpairs. It uses solving a correction equation to expand the search subspace, and finds approximate eigenpairs as best approximations in such search subspace.

### 3.1. Subspace Extraction

Let $\mathcal{X} \subseteq \mathbb{R}^m$ and $\mathcal{Y} \subseteq \mathbb{R}^n$ with $\dim(\mathcal{X}) = k$ and $\dim(\mathcal{Y}) = \ell$, respectively. As stated in [31], we call $\{\mathcal{X}, \mathcal{Y}\}$ a pair of defalting subspaces of CCGEP (4) if

$$C\mathcal{Y} \subseteq A\mathcal{X} \quad \text{and} \quad C^{\text{T}} \mathcal{X} \subseteq B\mathcal{Y}. \tag{13}$$

Let $X \in \mathbb{R}^{m \times k}$ and $Y \in \mathbb{R}^{n \times \ell}$ be $A$- and $B$-orthonormal basis matrices of the subspaces $\mathcal{X}$ and $\mathcal{Y}$, respectively, i.e.,

$$X^{\text{T}} A X = I_k \quad \text{and} \quad Y^{\text{T}} B Y = I_\ell.$$

The equality (13) implies that there exist $D_A \in \mathbb{R}^{k \times \ell}$ and $D_B \in \mathbb{R}^{\ell \times k}$ [32] (Equation (2.11)) such that

$$CY = AXD_A \quad \text{and} \quad C^T X = BYD_B. \tag{14}$$

They imply $D_A = X^T C Y = (Y^T C^T X)^T = D_B^T$. So (14) is equivalent to

$$CY = AXD_A \quad \text{and} \quad C^T X = BYD_A^T.$$

Now if $(\lambda, \hat{x}, \hat{y})$ is a singular triplet of $D_A$, then $(\lambda, z)$ gives an eigenpair of (4), where $z = [x^T, y^T]^T$, $x = X\hat{x}$ and $y = Y\hat{y}$. This is because

$$Cy = C(Y\hat{y}) = (CY)\hat{y} = AXD_A\hat{y} = \lambda AX\hat{x} = \lambda Ax.$$

and similarly $C^T x = \lambda By$. That means

$$\begin{bmatrix} 0 & C \\ C^T & 0 \end{bmatrix} \begin{bmatrix} x \\ y \end{bmatrix} = \lambda \begin{bmatrix} A & 0 \\ 0 & B \end{bmatrix} \begin{bmatrix} x \\ y \end{bmatrix}.$$

Hence, we have shown that a pair of deflating subspaces $\{\mathcal{X}, \mathcal{Y}\}$ can be used to recover those eigenpairs associated with the pair of deflating subspaces of CCGEP (4). In practice, pairs of exact deflating subspaces are usually not available, and one usually use Lanczos type methods [14] or Davidson type methods [15] to generate approximate ones, such as Krylov subspaces in Lanczos method [33]. Next, we will discuss how to extract best approximate eigenpairs from a given pair of approximate deflating subspaces.

In what follows, we consider the simple case: $k = \ell$. Suppose $\{\mathcal{U}, \mathcal{V}\}$ is an approximation of a pair of deflating subspaces $\{\mathcal{X}, \mathcal{Y}\}$ with $\dim(\mathcal{U}) = \dim(\mathcal{V}) = k$. Let $U \in \mathbb{R}^{m \times k}$ and $V \in \mathbb{R}^{n \times k}$ be the $A$- and $B$-orthonormal basis matrices of the subspaces $\mathcal{U}$ and $\mathcal{V}$, respectively. Denote $\theta_i$, $i = 1, 2, \ldots, k$ the singular values of $U^T C V$ in descending order with associated left and right singular vectors $\hat{u}_i$ and $\hat{v}_i$, respectively, i.e.,

$$(U^T C V)\hat{v}_i = \theta_i \hat{u}_i \quad \text{and} \quad (U^T C V)^T \hat{u}_i = \theta_i \hat{v}_i, \quad \text{for } 1 \leq i \leq k.$$

Even though $U$ and $V$ as $A$- and $B$-orthonormal basis matrices are not unique, these $\theta_i$ are. Motivated by the maximization principle in Theorem 1, we would seek the best approximations associated with the pair of approximate deflating subspaces $\{\mathcal{U}, \mathcal{V}\}$ to the eigenpairs $(\lambda_i, z_i)$ $(1 \leq i \leq j \leq k)$ in the sense of

$$\max \sum_{i=1}^{j} \sigma_i(\widetilde{U}_j^T C \widetilde{V}_j). \tag{15}$$

for any $\widetilde{U}_j \in \mathbb{R}^{m \times j}$ and $\widetilde{V}_j \in \mathbb{R}^{n \times j}$ satisfying $\text{span}(\widetilde{U}_j) \subseteq \mathcal{U}$, $\text{span}(\widetilde{V}_j) \subseteq \mathcal{V}$ and $\widetilde{U}_j^T A \widetilde{U}_j = \widetilde{V}_j^T B \widetilde{V}_j = I_j$. We claim that the quantity in (15) is given by $\sum_{i=1}^{j} \theta_i$. To see this, we notice that any $\widetilde{U}_j$ and $\widetilde{V}_j$ in (15) can be written as

$$\widetilde{U}_j = U\widehat{U}_j \quad \text{and} \quad \widetilde{V}_j = V\widehat{V}_j$$

for some $\widehat{U}_j \in \mathbb{R}^{k \times j}$ and $\widehat{V}_j \in \mathbb{R}^{k \times j}$ with $\widehat{U}_j^T \widehat{U}_j = \widehat{V}_j^T \widehat{V}_j = I_j$, and vice versa. Therefore the quantity in (15) is equal to

$$\max_{\widehat{U}_j^T \widehat{U}_j = \widehat{V}_j^T \widehat{V}_j = I_j} \sum_{i=1}^{j} \sigma_i(\widehat{U}^T U^T C V \widehat{V}),$$

which is $\sum_{i=1}^{j} \theta_i$ by the proposition of the singular value decomposition of $U^T C V$ [30]. The maximum is attended at $\widehat{U}_j = [\hat{u}_1, \hat{u}_2, \ldots, \hat{u}_j]$ and $\widehat{V}_j = [\hat{v}_1, \hat{v}_2, \ldots, \hat{v}_j]$. Therefore naturally, the best approximations to $(\lambda_i, z_i)$ $(1 \leq i \leq j)$ in the sense of (15) are given by

$$(\theta_i, \tilde{z}_i), \quad \text{where} \quad \tilde{z}_i = \begin{bmatrix} \tilde{x}_i \\ \tilde{y}_i \end{bmatrix}, \quad \tilde{x}_i = U\hat{u}_i \quad \text{and} \quad \tilde{y}_i = V\hat{v}_i. \tag{16}$$

Define the residual

$$r_i := K\tilde{z}_i - \theta_i M \tilde{z}_i = \begin{bmatrix} 0 & C \\ C^{\mathrm{T}} & 0 \end{bmatrix} \begin{bmatrix} \tilde{x}_i \\ \tilde{y}_i \end{bmatrix} - \theta_i \begin{bmatrix} A & 0 \\ 0 & B \end{bmatrix} \begin{bmatrix} \tilde{x}_i \\ \tilde{y}_i \end{bmatrix} = \begin{bmatrix} r_a^{(i)} \\ r_b^{(i)} \end{bmatrix}, \tag{17}$$

where $K$ and $M$ defined in (4), $r_a^{(i)} = C\tilde{y}_i - \theta_i A \tilde{x}_i$ and $r_b^{(i)} = C^{\mathrm{T}} \tilde{x}_i - \theta_i B \tilde{y}_i$. It is noted that

$$U^{\mathrm{T}} r_a^{(i)} = U^{\mathrm{T}} (C\tilde{y}_i - \theta_i A \tilde{x}_i) = U^{\mathrm{T}} CV\hat{v}_i - \theta_i U^{\mathrm{T}} A U \hat{u}_i = 0$$

and similarly $V^{\mathrm{T}} r_b^{(i)} = 0$. We summarize what we do in this subsection in the following theorem.

**Theorem 2.** *Suppose $\{\mathcal{U}, \mathcal{V}\}$ is a pair of approximate deflating subspaces with $\dim(\mathcal{U}) = \dim(\mathcal{V}) = k$. Let $U \in \mathbb{R}^{m \times k}$ and $V \in \mathbb{R}^{n \times k}$ be the A- and B-orthonormal basis matrices of the subspaces $\mathcal{U}$ and $\mathcal{V}$, respectively. Denote $\theta_i$, $i = 1, 2, \ldots, k$ the singular values of $U^{\mathrm{T}} CV$ in descending order. Then, for any $j \leq k$,*

$$\sum_{i=1}^{j} \theta_i = \max_{\substack{\mathrm{span}(\tilde{U}_j) \subseteq \mathcal{U}, \mathrm{span}(\tilde{V}_j) \subseteq \mathcal{V} \\ \tilde{U}_j^{\mathrm{T}} A \tilde{U}_j = \tilde{V}_j^{\mathrm{T}} B \tilde{V}_j = I_j}} \sum_{i=1}^{j} \sigma_i(\tilde{U}_j^{\mathrm{T}} C \tilde{V}_j),$$

*the best approximations to the eigenpairs $(\lambda_i, z_i)$ ($1 \leq i \leq j$) in the sense of (15) are $(\theta_i, \tilde{z}_i)$ ($1 \leq i \leq j$) given by (16), and the associated residuals defined in (17) admit $r_a^{(i)} \perp \mathcal{U}$ and $r_b^{(i)} \perp \mathcal{V}$.*

### 3.2. Correction Equation

In this subsection, we turn to construct a correction equation for a given eigenpair approximation. Suppose $(\theta, [\tilde{x}^{\mathrm{T}}, \tilde{y}^{\mathrm{T}}]^{\mathrm{T}})$ with $\tilde{x}^{\mathrm{T}} A \tilde{x} = \tilde{y}^{\mathrm{T}} B \tilde{y} = 1$ is an approximation of the eigenpair $(\lambda, [x^{\mathrm{T}}, y^{\mathrm{T}}]^{\mathrm{T}})$ of CCGEP (4), and $[r_a^{\mathrm{T}}, r_b^{\mathrm{T}}]^{\mathrm{T}}$ is the associated residual. We seek $A$- and $B$-orthogonal modifications of $\tilde{x}$ and $\tilde{y}$, respectively, such that

$$\begin{bmatrix} 0 & C \\ C^{\mathrm{T}} & 0 \end{bmatrix} \begin{bmatrix} \tilde{x} + s \\ \tilde{y} + t \end{bmatrix} = \lambda \begin{bmatrix} A & 0 \\ 0 & B \end{bmatrix} \begin{bmatrix} \tilde{x} + s \\ \tilde{y} + t \end{bmatrix}, \tag{18}$$

where $s \perp_A \tilde{x}$ and $t \perp_B \tilde{y}$. Then, by (18), we have

$$\begin{bmatrix} -\lambda A & C \\ C^{\mathrm{T}} & -\lambda B \end{bmatrix} \begin{bmatrix} s \\ t \end{bmatrix} = - \begin{bmatrix} r_a \\ r_b \end{bmatrix} + (\lambda - \theta) \begin{bmatrix} A & 0 \\ 0 & B \end{bmatrix} \begin{bmatrix} \tilde{x} \\ \tilde{y} \end{bmatrix}. \tag{19}$$

Notice that $r_a \perp \tilde{x}$ and $r_b \perp \tilde{y}$ by Theorem 2, which gives rise to

$$\begin{bmatrix} I_m - A\tilde{x}\tilde{x}^{\mathrm{T}} & 0 \\ 0 & I_n - B\tilde{y}\tilde{y}^{\mathrm{T}} \end{bmatrix} \begin{bmatrix} r_a \\ r_b \end{bmatrix} = \begin{bmatrix} r_a \\ r_b \end{bmatrix}, \quad \begin{bmatrix} I_m - A\tilde{x}\tilde{x}^{\mathrm{T}} & 0 \\ 0 & I_n - B\tilde{y}\tilde{y}^{\mathrm{T}} \end{bmatrix} \begin{bmatrix} A\tilde{x} \\ B\tilde{y} \end{bmatrix} = 0,$$

and

$$\begin{bmatrix} I_m - A\tilde{x}\tilde{x}^{\mathrm{T}} & 0 \\ 0 & I_n - B\tilde{y}\tilde{y}^{\mathrm{T}} \end{bmatrix} \begin{bmatrix} -\lambda A & C \\ C^{\mathrm{T}} & -\lambda B \end{bmatrix} \begin{bmatrix} s \\ t \end{bmatrix} = - \begin{bmatrix} r_a \\ r_b \end{bmatrix}. \tag{20}$$

Because $s \perp_A \tilde{x}$ and $t \perp_B \tilde{y}$, Equation (20) is rewritten as

$$\begin{bmatrix} I_m - A\tilde{x}\tilde{x}^{\mathrm{T}} & 0 \\ 0 & I_n - B\tilde{y}\tilde{y}^{\mathrm{T}} \end{bmatrix} \begin{bmatrix} -\lambda A & C \\ C^{\mathrm{T}} & -\lambda B \end{bmatrix} \begin{bmatrix} I_m - \tilde{x}\tilde{x}^{\mathrm{T}} A & 0 \\ 0 & I_n - \tilde{y}\tilde{y}^{\mathrm{T}} B \end{bmatrix} \begin{bmatrix} s \\ t \end{bmatrix} = - \begin{bmatrix} r_a \\ r_b \end{bmatrix}. \tag{21}$$

However, we do not know $\lambda$ here. It is natural that we use $\theta$ to replace $\lambda$ in (21) to get the final correction equation, i.e.,

$$
\begin{bmatrix} I_m - A\tilde{x}\tilde{x}^T & 0 \\ 0 & I_n - B\tilde{y}\tilde{y}^T \end{bmatrix} \begin{bmatrix} -\theta A & C \\ C^T & -\theta B \end{bmatrix} \begin{bmatrix} I_m - \tilde{x}\tilde{x}^T A & 0 \\ 0 & I_n - \tilde{y}\tilde{y}^T B \end{bmatrix} \begin{bmatrix} s \\ t \end{bmatrix} = - \begin{bmatrix} r_a \\ r_b \end{bmatrix}. \tag{22}
$$

We summarize what we have so far into Algorithm 1, and make a few comments on Algorithm 1.

(1) At step 2, $A$- and $B$-orthogonality procedures are applied to make sure $U^T A \tilde{t} = 0$ and $V^T B \tilde{s} = 0$.

(2) At step 7, in most cases, the correct equation is not necessity to solve exactly. Some steps of iterative methods for symmetric linear systems, such as linear conjugate gradient method (CG) [34] or the minimum residual method (MINRES) [35], are sufficient. Usually, more steps in solving the correction equation lead to fewer outer iterations. This will be shown in numerical examples.

(3) For the convergence test, we use the relative residual norms

$$
\eta(\theta_i, \tilde{z}_i) := \frac{\|r_a^{(i)}\|_1 + \|r_b^{(i)}\|_1}{(\|C\|_1 + \theta_i \|A\|_1)\|\tilde{x}_i\|_1 + (\|C\|_1 + \theta_i \|B\|_1)\|\tilde{y}_i\|_1} \tag{23}
$$

to determine if the approximate eigenparis $(\theta_i, \tilde{z}_i)$ has converged to a desired accuracy. In addition, in the practical implementation, once one or several of approximate eigenpairs converge to a preset accuracy, they should be deflated so that they will not be re-computed in the following iterations. Suppose $\lambda_i$ for $1 \le i \le j$, $X_j = [x_1, \dots, x_j]$ and $Y_j = [y_1, \dots, y_j]$ have been computed where $j \le k$. We can consider the generalized eigenvalue problem

$$
\widetilde{K}z = \lambda M z, \tag{24}
$$

where

$$
\widetilde{K} = \begin{bmatrix} I_m - AX_j X_j^T & 0 \\ 0 & I_n - BY_j Y_j^T \end{bmatrix} \begin{bmatrix} 0 & C \\ C^T & 0 \end{bmatrix} \begin{bmatrix} I_m - X_j X_j^T A & 0 \\ 0 & I_n - Y_j Y_j^T B \end{bmatrix}. \tag{25}
$$

By (11), it is clear that the eigenvalues of (24) consist of two groups. Those eigenvalues associated with the eigenvectors $[x_1^T, y_1^T]^T, \dots, [x_j^T, y_j^T]^T, [x_1^T, -y_1^T]^T, \dots, [x_j^T, -y_j^T]^T$ are shifted to zero and the others remain unchanged. Furthermore, for the correction equation, we find $s$ and $t$ subject to additional $A$- and $B$-orthogonality constrains for $s$ and $t$ against $X_j$ and $Y_j$, respectively. By a similar derivation of (22), the correction equation after deflation becomes

$$
\begin{bmatrix} I_m - A\tilde{x}\tilde{x}^T & 0 \\ 0 & I_n - B\tilde{y}\tilde{y}^T \end{bmatrix} \begin{bmatrix} -\theta_1 A & \widehat{C} \\ \widehat{C}^T & -\theta_1 B \end{bmatrix} \begin{bmatrix} I_m - \tilde{x}\tilde{x}^T A & 0 \\ 0 & I_n - \tilde{y}\tilde{y}^T B \end{bmatrix} \begin{bmatrix} s \\ t \end{bmatrix} = - \begin{bmatrix} r_a^{(1)} \\ r_b^{(1)} \end{bmatrix}, \tag{26}
$$

where $\widehat{C} = (I_m - AX_j X_j^T)C(I_n - Y_j Y_j^T B)$. Notice that $s \perp_A X_j$ and $t \perp_B Y_j$ mean $U \perp_A X_j$ and $V \perp_B Y_j$ in Algorithm 1, respectively. It follows that $U^T \widehat{C} V = U^T C V$.

(4) At step 5, LAPACK's routine xGESVD can be used to solve the singular value problem of $U^T C V$ because of its small size, where $U^T C V$ takes the following form:

$$U^{\mathrm{T}}CV = $$ 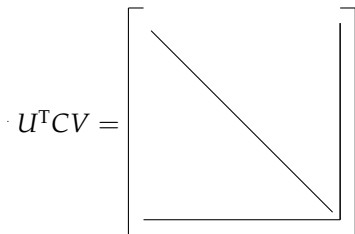

This form is preserved in the algorithm during refining the basis $U$ and $V$ at step 8. The new basis matrices $U\widehat{U}$ and $V\widehat{V}$ are reassigned to $U$ and $V$, respectively. Although a few extra costs are incurred, this refinement is necessary in order to have faster convergence for eigenvectors as stated in [36,37]. Furthermore, the restart is easily executed by keeping the first $s_{\min}$ columns of $U$ and $V$ when the dimension of the subspaces span$\{U\}$ and span$\{V\}$ exceeds $s_{\max}$. The restart technique appears at step 8 to keep the size of $U$, $V$ and $U^{\mathrm{T}}CV$ small. There are many ways to specify $s_{\max}$ and $s_{\min}$. In our numerical examples, we just simply take $s_{\max} = 3k$ and $s_{\min} = k$.

---

**Algorithm 1** Jacobi–Davidson method for canonical correlation analysis (JDCCA)

---

**Input:** Initial vectors $u_0, v_0, s = u_0, t = v_0$ and $V = U = [\,]$.
**Output:** Converged canonical weight vectors $\tilde{x}_i$ and $\tilde{y}_i$ for $i = 1, \dots, k$.
 1: **for** *iter* $= 1, 2, \dots,$until convergence **do**
 2:   $A$- and $B$-orthogonal $s$ and $t$ against $U$ and $V$, respectively, to obtain
      $\tilde{s}$ and $\tilde{t}$.
 3:   Compute $u = \tilde{s}/\|\tilde{s}\|_A$ and $v = \tilde{t}/\|\tilde{t}\|_B$. Let $U = [U, u]$ and $V = [V, v]$.
 4:   Update the corresponding column and row of $U^{\mathrm{T}}CV$.
 5:   Compute the singular value decomposition of $U^{\mathrm{T}}CV$, i.e., $U^{\mathrm{T}}CV = \widehat{U}\Theta\widehat{V}^{\mathrm{T}}$.
 6:   Compute the wanted approximate eigenpairs $(\theta_i, [\tilde{x}_i^{\mathrm{T}}, \tilde{y}_i^{\mathrm{T}}]^{\mathrm{T}})$ by (16) and the
      corresponding residuals $r_a^{(i)}$ and $r_b^{(i)}$.
 7:   Solve

$$\begin{bmatrix} I_m - A\tilde{x}_1\tilde{x}_1^{\mathrm{T}} & 0 \\ 0 & I_n - B\tilde{y}_1\tilde{y}_1^{\mathrm{T}} \end{bmatrix} \begin{bmatrix} -\theta_1 A & C \\ C^{\mathrm{T}} & -\theta_1 B \end{bmatrix} \begin{bmatrix} I_m - \tilde{x}_1\tilde{x}_1^{\mathrm{T}}A & 0 \\ 0 & I_n - \tilde{y}_1\tilde{y}_1^{\mathrm{T}}B \end{bmatrix} \begin{bmatrix} t \\ s \end{bmatrix} = - \begin{bmatrix} r_a^{(1)} \\ r_b^{(1)} \end{bmatrix}.$$

 8:   Update $U = U\widehat{U}$ and $V = V\widehat{V}$. If the dimension of $U$ and $V$ exceeds $s_{\max}$, then replace $U$ and $V$
      with $U_{(1:s_{\min})}$ and $V_{(1:s_{\min})}$ respectively.
 9: **end for**

---

### 3.3. Convergence

The convergence theories on the Jacobi–Davidson method for the eigenvalue and singular value problem are given in [15,38], respectively. Here we prove a similar convergence result for the Jacobi–Davidson method of CCGEP based on the following lemma. Specifically, if we solve the correction Equation (22) exactly, and then $\tilde{x}$ and $\tilde{y}$ are close enough to $x$ and $y$, respectively, it can be hoped that the approximate eigenvectors converge cubically. For the proof see Appendices A.2 and A.3.

**Lemma 1.** *Let $\lambda$ be a simple eigenvalue of CCGEP (4) with the corresponding eigenvector $[x^{\mathrm{T}}, y^{\mathrm{T}}]^{\mathrm{T}}$. Then the matrix*

$$G := \begin{bmatrix} I_m - Axx^{\mathrm{T}} & 0 \\ 0 & I_n - Byy^{\mathrm{T}} \end{bmatrix} \begin{bmatrix} -\lambda A & C \\ C^{\mathrm{T}} & -\lambda B \end{bmatrix} \begin{bmatrix} I_m - xx^{\mathrm{T}}A & 0 \\ 0 & I_n - yy^{\mathrm{T}}B \end{bmatrix}$$

*is a bijection from* span$(x)^{\perp_A} \times$ span$(y)^{\perp_B}$ *onto itself, where* span$(x)^{\perp_A}$ *and* span$(y)^{\perp_B}$ *are A- and B-orthogonal complementary spaces of* span$(x)$ *and* span$(y)$, *respectively.*

**Theorem 3.** *Assume the condition of Lemma 1, $\sin \angle_A(x, \tilde{x}) = \mathcal{O}(\varepsilon)$ and $\sin \angle_B(y, \tilde{y}) = \mathcal{O}(\varepsilon)$. Let $[s^T, t^T]^T$ be the exact solution of the correction Equation (22). Then,*

$$|\sin \angle_A(x, \tilde{x} + s)| = \mathcal{O}(\varepsilon^3) \quad and \quad |\sin \angle_B(y, \tilde{y} + t)| = \mathcal{O}(\varepsilon^3). \tag{27}$$

## 4. Numerical Examples

In this section, we present some numerical examples to illustrate the effectiveness of Algorithm 1. Our goal is to compute the first few canonical weight vectors. A computed approximate eigenpair $(\theta_i, \tilde{z}_i)$ is considered converged when its relative residual norm

$$\eta(\theta_i, \tilde{z}_i) = \frac{\|r_a^{(i)}\|_1 + \|r_b^{(i)}\|_1}{(\|C\|_1 + \theta_i\|A\|_1)\|\tilde{x}_i\|_1 + (\|C\|_1 + \theta_i\|B\|_1)\|\tilde{y}_i\|_1} \leq 10^{-8}. \tag{28}$$

All the experiments in this paper are executed on a Ubuntu 12.04 (64 bit) Desktop-Intel(R) Core(TM) i7-6700 CPU@3.40 GHz, 32 GB of RAM using MATLAB 2010a with machine epsilon $2.22 \times 10^{-16}$ in double-precision floating point arithmetic.

**Example 1.** *We first examine Theorem 3 by using two pairs of data matrices $S_a$ and $S_b$ which come from a publicly available handwritten numerals dataset (https://archive.ics.uci.edu/ml/datasets/Multiple+Features). It consists of features handwritten numerals ('0'–'9') and each digit has 200 patterns. Each pattern is represented by six different feature sets, i.e., Fou, Fac, Kar, Pix, Zer and Mor. Two pairs of feature sets Fou-Zer and Pix-Fou are chosen for $S_a$ and $S_b$, respectively, such that $S_a \in \mathbb{R}^{76 \times d}$ and $S_b \in \mathbb{R}^{47 \times d}$ in Fou-Zer, and $S_a \in \mathbb{R}^{240 \times d}$ and $S_b \in \mathbb{R}^{76 \times d}$ in Pix-Fou with $d = 2000$. To make the numerical example repeatable, the initial vectors are set to be*

$$u_0 = x_1 + 10^{-3} \times \text{ones}(m, 1) \quad and \quad v_0 = y_1 + 10^{-3} \times \text{ones}(n, 1)$$

*where m and n are the dimension of $S_a$ and $S_b$, respectively, ones is MATLAB built-in function, and $[x_1^T, y_1^T]^T$ is computed by MATLAB's function eig on (4) and considered to be the "exact" eigenvector for testing purposes. The corrected Equation (22) in Algorithm 1 is solved by direct methods, such as Gaussian elimination, and the solution $[s^T, t^T]^T$ by such methods is regarded as "exactly" in this example. Figure 1 plots $\sin \angle_A(x_1, \tilde{x}_1)$ and $\sin \angle_B(y_1, \tilde{y}_1)$ in the first three iterations of Algorithm 1 for computing first canonical weight vector of Fou-Zer and Pix-Fou. It is clearly shown by Figure 1 that the convergence of Algorithm 1 is very fast when the initial vectors are enough close to the exact vectors, and the cubical convergence of Algorithm 1 appears in the third iteration.*

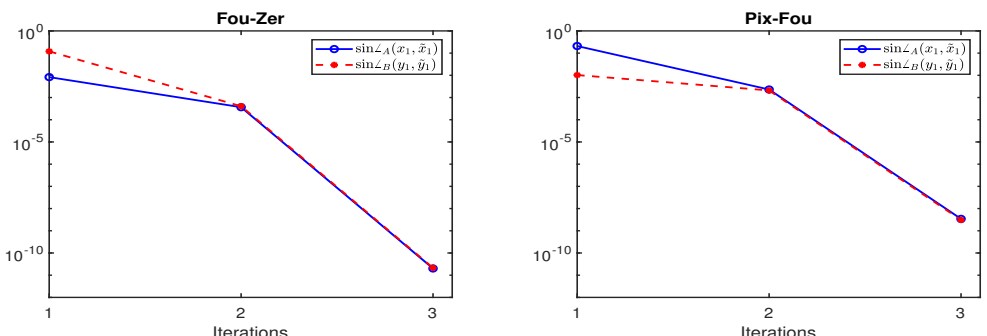

**Figure 1.** Convergence behavior of Algorithm 1 for computing the first canonical weight vector of Fou-Zer and Pix-Fou.

**Example 2.** *As stated in Algorithm 1, the implementation of JDCCA involves solving the correction Equation (22) in every step. Direct solvers mentioned in Example 1 referring to $\mathcal{O}\left((m + n)^3\right)$ operations are prohibitively expensive in solving large-scale sparse linear systems. In such a case, iterative methods, such as MINRES method which is simply GMRES [39] applied to symmetric linear systems, are usually preferred. In this example, we report the effect of the number of steps in the solution of the correction equation,*

denoted by $n_g$, on the total number of matrix-vector products (denoted by "#mvp"), outer iteration number (denoted by "#iter"), and CPU time in seconds for Algorithm 1 to compute the first 10 canonical weight vectors of the test problems appeared in Table 1. Table 1 presents three face datasets, i.e., ORL (*https://www.cl.cam. ac.uk/research/dtg/attarchive/facedatabase.html*,) FERET (*http://www.nist.gov/itl/iad/ig/colorferet.cfm*) and Yale (*https://computervisiononline.com/dataset/1105138686*) datasets. The ORL database contains 400 face images of 40 distinct persons. For each individual, there are 10 different gray scale images with $92 \times 112$ pixels. These images are collected by volunteers at different times, different lighting and different facial expressions (blinking or closed eyes, smiling or no-smiling, wearing glasses or no-glasses). In order to apply CCA, as numerical experiments in [40], the ORL dataset is partitioned into two groups. We select the first five images per individual as the first view to generate the data matrix $S_a$, while the remaining for $S_b$. Similarly, we get data matrices $S_a$ and $S_b$ for the FERET and Yale datasets. The numbers of row and column of $S_a$ and $S_b$ are detailed in Table 1.

**Table 1.** Test problems.

| Problems | ORL | FERET | Yale |
|:---:|:---:|:---:|:---:|
| $m$ | 10,304 | 6400 | 10,000 |
| $n$ | 10,304 | 6400 | 10,000 |
| $d$ | 200 | 600 | 75 |

In this example, we set the initial vectors $u_0 = \text{ones}(m, 1)$ and $v_0 = \text{ones}(n, 1)$ with $s_{\max} = 30$ and $s_{\min} = 10$ for restarting and simply take regularization parameter $\kappa_a = \kappa_b = 10^{-4}$. We let MINRES steps $n_g$ vary from to 5 to 40, and collect the numerical results in Figure 2. As expected, the number of total outer iterations decreases as $n_g$ increases, while the total number of matrix-vector products does not change monotonically with $n_g$. It depends on the degree of reduction of outer iterations by the increasing of $n_g$. In addition, it is shown by Figure 2 that the total #mvp is not a unique deciding factor on the total CPU time. When $n_g$ is larger, the significantly reduced #iter leads to smaller total CPU time. For these three test examples, the MINRES steps $n_g$ around 15 to 25 appear to be cost-effective, further increasing $n_g$ over 40 usually does not have significant effect. The least efficient case is when $n_g$ is too small.

**Example 3.** *In this example, we compare Algorithm 1, i.e., JDCCA, with Jacobi–Davidson QZ type method [41] (JDQZ) for the large scale symmetric positive definite pencil $\{K, M\}$ defined in (4) to compute the first 10 canonical weight vectors of the test problems appeared in Table 1 with MINRES steps $n_g = 20$. We take $u_0 = \text{ones}(m, 1)$ and $v_0 = \text{ones}(n, 1)$ in Algorithm 1 and the initial vector $\text{ones}(m + n, 1)$ for the JDQZ algorithm, and compute the same relative residual norms $\eta(\theta_i, \tilde{z}_i)$. The corresponding numerical results are plotted in Figure 3. For these three test problems, it is suggested by Figure 3 that Algorithm 1 always outperforms the JDQZ algorithm. Other experiments that we tested with different test problems and MINRES steps not reported here also illustrate our points.*

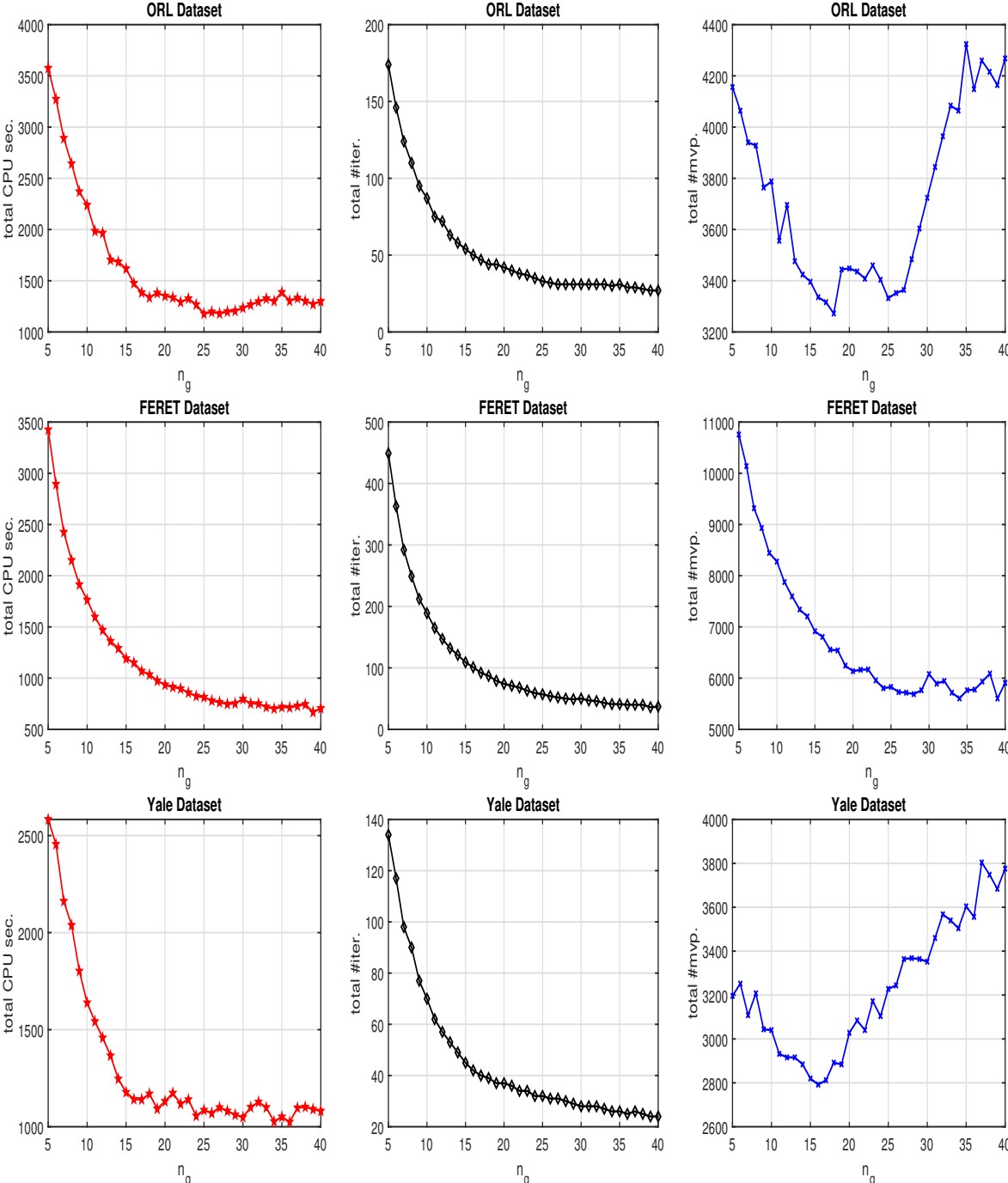

**Figure 2.** Cost in computing the first 10 canonical weight vectors of ORL (**top**), FERET (**middle**) and Yale (**bottom**) datasets with MINRES steps for the correction equation varying from 5 to 40.

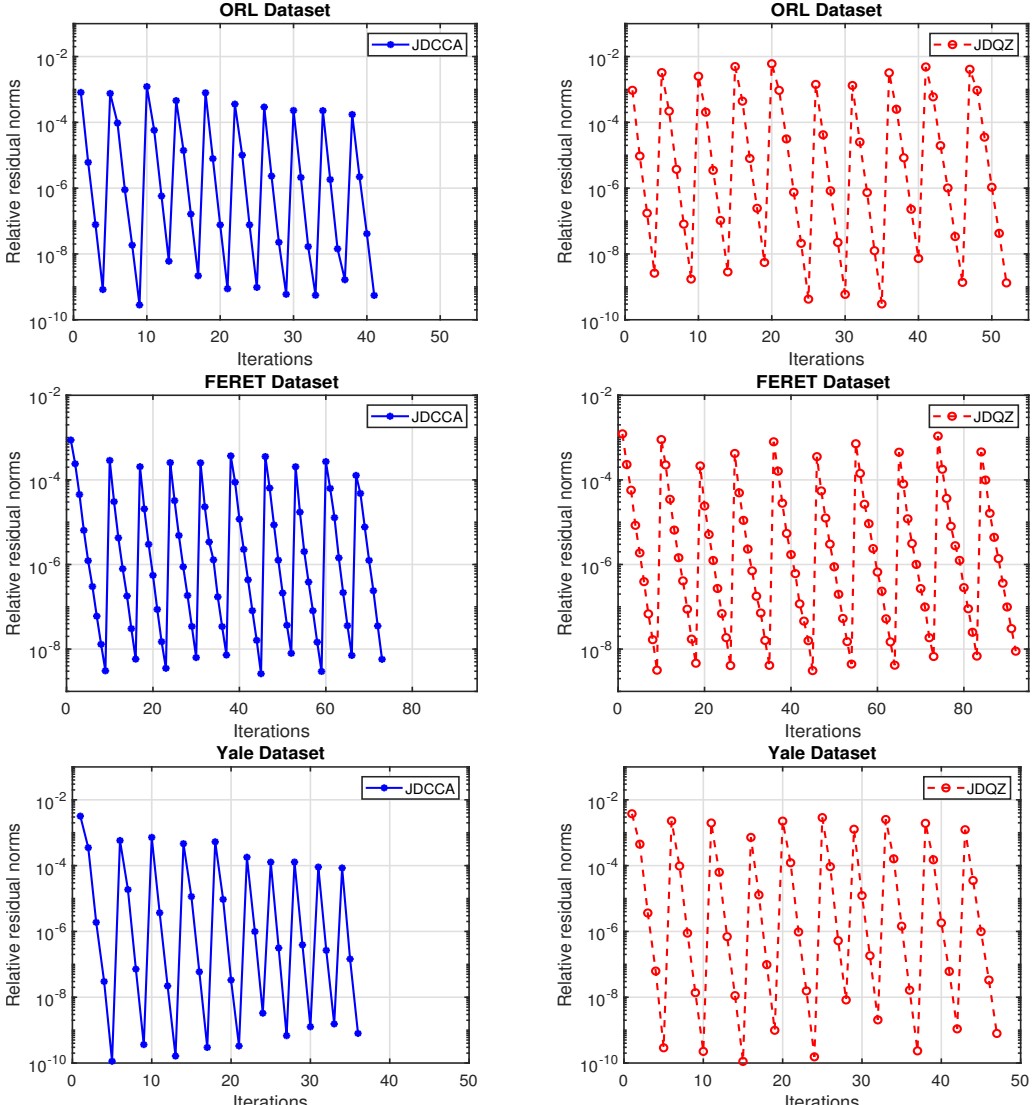

**Figure 3.** Convergence behavior of JDCCA and JDQZ for computation of the first 10 canonical weight vectors of ORL (**top**), FERET (**middle**) and Yale (**bottom**) datasets with MINRES step $n_g = 20$.

## 5. Conclusions

To analyze the correlations between two data sets, several numerical algorithms have been available to find the canonical correlations and the associated canonical weight vectors; however, there is very little discussion of the large scale sparse and structured matrix cases in the literature. In this paper, a Jacobi–Davidson type method, i.e., Algorithm 1, is presented for large scale canonical correlation analysis by computing a small portion of eigenpairs of the canonical correlation generalized eigenvalue problem (4). The theoretical result is established in Theorem 3 to demonstrate that the cubic convergence of the approximate eigenvector if the correction equation is solved exactly and the approximate eigenvector of the previous step is close enough to the exact one. The corresponding numerical results are presented to confirm the effectiveness of asymptotic convergence rate provided by Theorem 3, and to demonstrate that Algorithm 1 performs far superior to the JDQZ method for the large scale symmetric positive definite pencil $\{K, M\}$.

Notice that the main computational tasks in every iteration of Algorithm 1 consist of solving the correction Equation (22). In our numerical example, we only focus on the plain version of MINRES, i.e., without considering any preconditioner. However, it is not hard to notice that incorporating a preconditioner presents no difficulty and can promote the numerical performance if

the preconditioner is available. In addition, from the point of view that multi-set canonical correlation analysis (MCCA) [42] proposed to analyze linear relationships among more than two data sets can be equivalently transformed to the following generalized eigenvalue problem

$$
\begin{bmatrix}
0 & C_{12} & \dots & C_{1t} \\
C_{21} & 0 & \dots & C_{2t} \\
\vdots & \vdots & \vdots & \vdots \\
C_{t1} & C_{t2} & \dots & 0
\end{bmatrix}
\begin{bmatrix}
x_1 \\ x_2 \\ \vdots \\ x_t
\end{bmatrix}
= \lambda
\begin{bmatrix}
C_{11} & 0 & \dots & 0 \\
0 & C_{22} & \dots & 0 \\
\vdots & \vdots & \vdots & \vdots \\
0 & 0 & \dots & C_{tt}
\end{bmatrix}
\begin{bmatrix}
x_1 \\ x_2 \\ \vdots \\ x_t
\end{bmatrix},
$$

where $C_{ij} = S_i S_j^{\mathrm{T}}$ and $S_i$ is the data matrix, the development of efficient Jacobi–Davidson methods for treating such large scale MCCA will be a subject of our future study.

**Author Contributions:** Writing—original draft, Z.T.; Writing—review and editing, X.Z. and Z.T. All authors have read and agreed to the published version of the manuscript.

**Funding:** This work is supported in part by National Natural Science Foundation of China NSFC-11601081 and the research fund for distinguished young scholars of Fujian Agriculture and Forestry University No. xjq201727.

**Conflicts of Interest:** The authors declare no conflict of interest.

## Appendix A

*Appendix A.1*

**Proof of Theorem 1.** To prove (12), for any $U \in \mathbb{R}^{m \times k}$ and $V \in \mathbb{R}^{n \times \ell}$ satisfying $U^{\mathrm{T}} A U = I_k$ and $V^{\mathrm{T}} B V = I_\ell$, respectively, we first consider the augmented matrices of $\widetilde{C}$ and $U^{\mathrm{T}} C V$, i.e.,

$$
\begin{bmatrix} 0 & \widetilde{C} \\ \widetilde{C}^{\mathrm{T}} & 0 \end{bmatrix}
\quad \text{and} \quad
\begin{bmatrix} 0 & U^{\mathrm{T}} C V \\ V^{\mathrm{T}} C^{\mathrm{T}} U & 0 \end{bmatrix},
$$

whose eigenvalues $\pm \lambda_i$ for $i = 1, \dots, n$ plus $m - n$ zeros and $\sigma_i(U^{\mathrm{T}} C V)$ for $i = 1, \dots, \min\{k, \ell\}$ plus $k + \ell - 2\min\{k, \ell\}$, respectively. Notice that

$$
\begin{bmatrix} 0 & U^{\mathrm{T}} C V \\ V^{\mathrm{T}} C^{\mathrm{T}} U & 0 \end{bmatrix}
=
\begin{bmatrix} U^{\mathrm{T}} R_{\mathrm{A}}^{\mathrm{T}} & 0 \\ 0 & V^{\mathrm{T}} R_{\mathrm{B}}^{\mathrm{T}} \end{bmatrix}
\begin{bmatrix} 0 & \widetilde{C} \\ \widetilde{C}^{\mathrm{T}} & 0 \end{bmatrix}
\begin{bmatrix} R_{\mathrm{A}} U & 0 \\ 0 & R_{\mathrm{B}} V \end{bmatrix},
$$

where $R_{\mathrm{A}}$ and $R_{\mathrm{B}}$ are defined in (5), and $R_{\mathrm{A}} U$ and $R_{\mathrm{B}} V$ satisfy $(R_{\mathrm{A}} U)^{\mathrm{T}} R_{\mathrm{A}} U = U^{\mathrm{T}} A U = I_k$ and $(R_{\mathrm{B}} V)^{\mathrm{T}} R_{\mathrm{B}} V = V^{\mathrm{T}} B V = I_\ell$, respectively. Hence, apply Cauchy's interlacing inequalities [30] (Corollary 4.4) for the symmetric eigenvalue problem to the matrices $\begin{bmatrix} 0 & \widetilde{C} \\ \widetilde{C}^{\mathrm{T}} & 0 \end{bmatrix}$ and $\begin{bmatrix} 0 & U^{\mathrm{T}} C V \\ V^{\mathrm{T}} C^{\mathrm{T}} U & 0 \end{bmatrix}$, to get $\lambda_i \geq \sigma_i(U^{\mathrm{T}} C V)$ for $1 \leq i \leq \min\{k, \ell\}$ and consequently

$$
\sum_{i=1}^{\min\{k,\ell\}} \lambda_i \geq \max \sum_{i=1}^{\min\{k,\ell\}} \sigma_i(U^{\mathrm{T}} C V) \tag{A1}
$$

for any $U \in \mathbb{R}^{m \times k}$ and $V \in \mathbb{R}^{n \times \ell}$ such that $U^{\mathrm{T}} A U = I_k$ and $V^{\mathrm{T}} B V = I_\ell$.

On the other hand, let $U = [x_1, x_2, \dots, x_k]$ and $V = [y_1, y_2, \dots, y_\ell]$ where $x_i$ and $y_i$ are defined in (10). Then, by (11), we have $U^{\mathrm{T}} A U = I_k$ and $V^{\mathrm{T}} B V = I_\ell$. Furthermore,

$$
U^{\mathrm{T}} C V = [p_1, p_2, \dots, p_k]^{\mathrm{T}} \widetilde{C} [q_1, q_2, \dots, q_\ell] = \mathrm{diag}(\lambda_1, \dots, \lambda_{\min\{k,\ell\}}) \in \mathbb{R}^{k \times \ell}
$$

which to give $\sigma_i(U^{\mathrm{T}} C V) = \lambda_i$ for $1 \leq i \leq \min\{k, \ell\}$ and thus

$$\sum_{i=1}^{\min\{k,\ell\}} \lambda_i \le \max_{U^\mathsf{T} A U = I_k,\, V^\mathsf{T} B V = I_\ell} \sum_{i=1}^{\min\{k,\ell\}} \sigma_i(U^\mathsf{T} C V). \tag{A2}$$

Equation (12) is a consequence of (A1) and (A2). □

*Appendix A.2*

**Proof of Lemma 1.** Let $[w_1^\mathsf{T}, w_2^\mathsf{T}]^\mathsf{T} \in \operatorname{span}(x)^{\perp_A} \times \operatorname{span}(y)^{\perp_B}$ and it satisfies $G[w_1^\mathsf{T}, w_2^\mathsf{T}]^\mathsf{T} = 0$. We will prove $[w_1^\mathsf{T}, w_2^\mathsf{T}]^\mathsf{T} = 0$. Since

$$G \begin{bmatrix} w_1 \\ w_2 \end{bmatrix} = \begin{bmatrix} I_m - Axx^\mathsf{T} & 0 \\ 0 & I_n - Byy^\mathsf{T} \end{bmatrix} \begin{bmatrix} -\lambda A & C \\ C^\mathsf{T} & -\lambda B \end{bmatrix} \begin{bmatrix} w_1 \\ w_2 \end{bmatrix},$$

then we have

$$\begin{bmatrix} -\lambda A & C \\ C^\mathsf{T} & -\lambda B \end{bmatrix} \begin{bmatrix} w_1 \\ w_2 \end{bmatrix} = \begin{bmatrix} \gamma_1 Ax \\ \gamma_2 By \end{bmatrix},$$

where $\gamma_1 = x^\mathsf{T}(Cw_2 - \lambda Aw_1)$ and $\gamma_2 = y^\mathsf{T}(C^\mathsf{T} w_1 - \lambda Bw_2)$, which leads to

$$\begin{cases} Cw_2 = \lambda Aw_1 + \gamma_1 Ax, \\ C^\mathsf{T} w_1 = \lambda Bw_2 + \gamma_2 By, \end{cases} \Rightarrow \begin{cases} Cw_2 = \lambda R_A^\mathsf{T} R_A w_1 + \gamma_1 R_A^\mathsf{T} R_A x, \\ C^\mathsf{T} w_1 = \lambda R_B^\mathsf{T} R_B w_2 + \gamma_2 R_B^\mathsf{T} R_B y. \end{cases} \tag{A3}$$

Let $\tilde{w}_1 = R_A w_1$ and $\tilde{w}_2 = R_B w_2$. Then the equality (A3) can be rewritten as

$$\begin{cases} \widetilde{C}\tilde{w}_2 = \lambda \tilde{w}_1 + \gamma_1 p, \\ \widetilde{C}^\mathsf{T} \tilde{w}_1 = \lambda \tilde{w}_2 + \gamma_2 q, \end{cases} \tag{A4}$$

where $\widetilde{C}$, $p$ and $q$ are defined in (7). Multiply the first and second equations of (A4) by $\widetilde{C}^\mathsf{T}$ and $\widetilde{C}$ from left, respectively, to get

$$\begin{cases} \widetilde{C}^\mathsf{T} \widetilde{C} \tilde{w}_2 = \lambda \widetilde{C}^\mathsf{T} \tilde{w}_1 + \gamma_1 \widetilde{C}^\mathsf{T} p = \lambda^2 \tilde{w}_2 + \lambda \gamma_2 q + \gamma_1 \lambda q, \\ \widetilde{C}\widetilde{C}^\mathsf{T} \tilde{w}_1 = \lambda \widetilde{C} \tilde{w}_2 + \gamma_2 \widetilde{C} q = \lambda^2 \tilde{w}_1 + \lambda \gamma_1 p + \gamma_1 \lambda p, \end{cases}$$

$$\Rightarrow \begin{cases} (\widetilde{C}^\mathsf{T} \widetilde{C} - \lambda^2 I_n)\tilde{w}_2 = (\lambda \gamma_2 + \lambda \gamma_1)q, \\ (\widetilde{C}\widetilde{C}^\mathsf{T} - \lambda^2 I_m)\tilde{w}_1 = (\lambda \gamma_1 + \lambda \gamma_1)p. \end{cases}$$

Therefore, both $\tilde{w}_1$ and $p$ belong to the kernel of $(\widetilde{C}\widetilde{C}^\mathsf{T} - \lambda^2 I_m)^2$, and both $\tilde{w}_2$ and $q$ belong to the kernel of $(\widetilde{C}^\mathsf{T} \widetilde{C} - \lambda^2 I_n)^2$. The simplicity of $\lambda$ implies $\tilde{w}_1$ and $\tilde{w}_2$ are multiples of $p$ and $q$, respectively. Since $w_1 \in \operatorname{span}(x)^{\perp_A}$ and $w_2 \in \operatorname{span}(y)^{\perp_B}$, we have $\tilde{w}_1 \in \operatorname{span}(p)^\perp$ and $\tilde{w}_2 \in \operatorname{span}(q)^\perp$, which means $\tilde{w}_1 = \tilde{w}_2 = 0$. Therefore, $w_1 = w_2 = 0$. The bijectivity follow from comparing dimensions. □

*Appendix A.3*

**Proof of Theorem 3.** Let

$$F := \begin{bmatrix} I_m - A\tilde{x}\tilde{x}^\mathsf{T} & 0 \\ 0 & I_n - B\tilde{y}\tilde{y}^\mathsf{T} \end{bmatrix}.$$

Then the correction equation is

$$F \begin{bmatrix} -\theta A & C \\ C^\mathsf{T} & -\theta B \end{bmatrix} F^\mathsf{T} \begin{bmatrix} s \\ t \end{bmatrix} = -\begin{bmatrix} r_a \\ r_b \end{bmatrix}. \tag{A5}$$

Since $\|\tilde{x}\|_A = \|x\|_A = 1$, there exists $f \perp_A \tilde{x}$ such that $x = \alpha\tilde{x} + f$ where $\alpha^2 + \|f\|_A^2 = 1$. It follows that $\frac{x}{\alpha} = \tilde{x} + \tilde{f}$ where $\tilde{f} = \frac{f}{\alpha}$ and $\|\tilde{f}\|_A = \tan \angle_A(x, \tilde{x}) = \mathcal{O}(\varepsilon)$. Similarly, there are $\tilde{g} \perp_B \tilde{y}$ and a scalar $\beta$ such that $\frac{y}{\beta} = \tilde{y} + \tilde{g}$ where $\|\tilde{g}\|_B = \tan \angle_B(y, \tilde{y}) = \mathcal{O}(\varepsilon)$. It is noted that

$$
\begin{aligned}
0 &= \begin{bmatrix} -\lambda A & C \\ C^{\mathsf{T}} & -\lambda B \end{bmatrix} \begin{bmatrix} x \\ y \end{bmatrix} = \begin{bmatrix} -\lambda\alpha A & \beta C \\ \alpha C^{\mathsf{T}} & -\lambda\beta B \end{bmatrix} \begin{bmatrix} \frac{x}{\alpha} \\ \frac{y}{\beta} \end{bmatrix} \\
&= \left( \begin{bmatrix} -\theta A & C \\ C^{\mathsf{T}} & -\theta B \end{bmatrix} + \begin{bmatrix} (\theta - \lambda\alpha)A & (\beta - 1)C \\ (\alpha - 1)C^{\mathsf{T}} & (\theta - \lambda\beta)B \end{bmatrix} \right) \begin{bmatrix} \frac{x}{\alpha} \\ \frac{y}{\beta} \end{bmatrix} \\
&= \begin{bmatrix} -\theta A & C \\ C^{\mathsf{T}} & -\theta B \end{bmatrix} \begin{bmatrix} \frac{x}{\alpha} \\ \frac{y}{\beta} \end{bmatrix} - \begin{bmatrix} \omega_1 A x \\ \omega_2 B y \end{bmatrix},
\end{aligned}
\tag{A6}
$$

where $\omega_1 = \frac{\lambda\alpha - \theta}{\alpha} + \frac{\lambda(1-\beta)}{\beta}$ and $\omega_2 = \frac{\lambda\beta - \theta}{\beta} + \frac{\lambda(1-\alpha)}{\alpha}$. Since $\frac{x}{\alpha} = \tilde{x} + \tilde{f}$ and $\frac{y}{\beta} = \tilde{y} + \tilde{g}$, the equality (A6) leads to

$$
\begin{aligned}
\begin{bmatrix} -\theta A & C \\ C^{\mathsf{T}} & -\theta B \end{bmatrix} \begin{bmatrix} \tilde{f} \\ \tilde{g} \end{bmatrix} &= - \begin{bmatrix} -\theta A & C \\ C^{\mathsf{T}} & -\theta B \end{bmatrix} \begin{bmatrix} \tilde{x} \\ \tilde{y} \end{bmatrix} + \begin{bmatrix} \omega_1 A x \\ \omega_2 B y \end{bmatrix} \\
&= - \begin{bmatrix} r_a \\ r_b \end{bmatrix} + \begin{bmatrix} \omega_1\alpha A(\tilde{x} + \tilde{f}) \\ \omega_2\beta B(\tilde{y} + \tilde{g}) \end{bmatrix}.
\end{aligned}
\tag{A7}
$$

It is noted that $F \begin{bmatrix} r_a \\ r_b \end{bmatrix} = \begin{bmatrix} r_a \\ r_b \end{bmatrix}$, $F \begin{bmatrix} A\tilde{x} \\ B\tilde{y} \end{bmatrix} = 0$, $F \begin{bmatrix} A\tilde{f} \\ B\tilde{g} \end{bmatrix} = \begin{bmatrix} A\tilde{f} \\ B\tilde{g} \end{bmatrix}$ and $F^{\mathsf{T}} \begin{bmatrix} \tilde{f} \\ \tilde{g} \end{bmatrix} = \begin{bmatrix} \tilde{f} \\ \tilde{g} \end{bmatrix}$. Then, we have by (A7)

$$
\begin{aligned}
F \begin{bmatrix} -\theta A & C \\ C^{\mathsf{T}} & -\theta B \end{bmatrix} F^{\mathsf{T}} \begin{bmatrix} \tilde{f} \\ \tilde{g} \end{bmatrix} &= -F \begin{bmatrix} r_a \\ r_b \end{bmatrix} + F \begin{bmatrix} \omega_1\alpha A\tilde{f} \\ \omega_2\beta B\tilde{g} \end{bmatrix} \\
&= - \begin{bmatrix} r_a \\ r_b \end{bmatrix} + \begin{bmatrix} \omega_1\alpha A\tilde{f} \\ \omega_2\beta B\tilde{g} \end{bmatrix}.
\end{aligned}
\tag{A8}
$$

Together (A5) with (A8) to get

$$
F \begin{bmatrix} -\theta A & C \\ C^{\mathsf{T}} & -\theta B \end{bmatrix} F^{\mathsf{T}} \begin{bmatrix} \tilde{f} - s \\ \tilde{g} - t \end{bmatrix} = \begin{bmatrix} \omega_1\alpha A\tilde{f} \\ \omega_2\beta B\tilde{g} \end{bmatrix}.
\tag{A9}
$$

In addition, since $r_a \perp \tilde{x}$ and $r_b \perp \tilde{y}$, multiplying (A7) on the left by $\begin{bmatrix} \tilde{x} & 0 \\ 0 & \tilde{y} \end{bmatrix}^{\mathsf{T}}$ leads to

$$
\begin{aligned}
\begin{bmatrix} \omega_1\alpha \\ \omega_2\beta \end{bmatrix} &= \begin{bmatrix} (\tilde{x}^{\mathsf{T}} A\tilde{x})^{-1} & \\ & (\tilde{y}^{\mathsf{T}} B\tilde{y})^{-1} \end{bmatrix} \begin{bmatrix} -\theta\tilde{x}^{\mathsf{T}} A & \tilde{x}^{\mathsf{T}} C \\ \tilde{y}^{\mathsf{T}} C^{\mathsf{T}} & -\theta\tilde{y}^{\mathsf{T}} B \end{bmatrix} \begin{bmatrix} \tilde{f} \\ \tilde{g} \end{bmatrix} \\
&= \begin{bmatrix} -\theta\tilde{x}^{\mathsf{T}} A\tilde{f} + \tilde{x}^{\mathsf{T}} C\tilde{g} \\ \tilde{y}^{\mathsf{T}} C^{\mathsf{T}}\tilde{f} - \theta\tilde{y}^{\mathsf{T}} B\tilde{g} \end{bmatrix} \qquad \text{(by } \tilde{x}^{\mathsf{T}} A\tilde{x} = \tilde{y}^{\mathsf{T}} B\tilde{y} = 1\text{)} \\
&= \begin{bmatrix} \tilde{x}^{\mathsf{T}} C\tilde{g} \\ \tilde{y}^{\mathsf{T}} C^{\mathsf{T}}\tilde{f} \end{bmatrix} = \begin{bmatrix} (\frac{x}{\alpha} - \tilde{f})^{\mathsf{T}} C\tilde{g} \\ (\frac{y}{\beta} - \tilde{g})^{\mathsf{T}} C^{\mathsf{T}}\tilde{f} \end{bmatrix} \\
&= \begin{bmatrix} (\frac{x}{\alpha})^{\mathsf{T}} C\tilde{g} - \tilde{f}^{\mathsf{T}} C\tilde{g} \\ (\frac{y}{\beta})^{\mathsf{T}} C^{\mathsf{T}}\tilde{f} - \tilde{g}^{\mathsf{T}} C^{\mathsf{T}}\tilde{f} \end{bmatrix} \\
&= \begin{bmatrix} \frac{\lambda\beta}{\alpha}\tilde{g}^{\mathsf{T}} B\tilde{g} - \tilde{f}^{\mathsf{T}} C\tilde{g} \\ \frac{\lambda\alpha}{\beta}\tilde{f}^{\mathsf{T}} A\tilde{f} - \tilde{g}^{\mathsf{T}} C^{\mathsf{T}}\tilde{f} \end{bmatrix}.
\end{aligned}
\tag{A10}
$$

By Lemma 1, when $\tilde{x}$ and $\tilde{y}$ are close enough to $x$ and $y$, respectively, we see that $F \begin{bmatrix} -\theta A & C \\ C^{\mathrm{T}} & -\theta B \end{bmatrix} F^{\mathrm{T}}$ is invertible. It follows by (A9) that

$$\begin{bmatrix} \tilde{f} - s \\ \tilde{g} - t \end{bmatrix} = \left( F \begin{bmatrix} -\theta A & C \\ C^{\mathrm{T}} & -\theta B \end{bmatrix} F^{\mathrm{T}} \right)^{-1} \begin{bmatrix} \omega_1 \alpha A \tilde{f} \\ \omega_2 \beta B \tilde{g} \end{bmatrix} \quad \Rightarrow \quad \left\| \begin{bmatrix} \tilde{f} - s \\ \tilde{g} - t \end{bmatrix} \right\|_M = \mathcal{O}\left(\varepsilon^3\right).$$

The last equality holds because of $\|\tilde{f}\|_A = \mathcal{O}(\varepsilon)$, $\|\tilde{g}\|_B = \mathcal{O}(\varepsilon)$ and (A10), which means $\|\tilde{f} - s\|_A = \mathcal{O}\left(\varepsilon^3\right)$ and $\|\tilde{g} - t\|_B = \mathcal{O}\left(\varepsilon^3\right)$. Therefore,

$$\begin{aligned} |\sin \angle_A(x, \tilde{x} + s)| &= |\sin \angle_A(x, \tfrac{x}{\alpha} + s - \tilde{f})| \\ &= \frac{\|X^\perp A(\tfrac{x}{\alpha} + s - \tilde{f})\|_2}{\|\tilde{x} + s\|_A} = \frac{\|X^\perp A(\tilde{f} - s)\|_2}{\|\tilde{x} + s\|_A} \\ &\leq \frac{\|\tilde{f} - s\|_A}{\|\tilde{x} + s\|_A} = \mathcal{O}\left(\varepsilon^3\right), \end{aligned}$$

where $X^\perp = [x_2, \ldots, x_m]$. Similarly, we have $|\sin \angle_B(y, \tilde{y} + t)| = \mathcal{O}\left(\varepsilon^3\right)$. □

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
