# Peer review of "A Jacobi–Davidson Method for Large Scale Canonical Correlation Analysis"

_algorithms, doi:10.3390/a13090229_

Round 1
Reviewer 1 Report
Dear Editor,
It seems the result is new and the proofs are okay.
More precisely, in this paper, the authors considered a Jacobi-Davidson type algorithm to evauate the canonical weight vectors by transforming it into the canonical correlation generalized eigenvalue problem.
For the accuracy of the observed approximation of the canonical weight vectors, convergence results are constructed.
In addition, some numerical examples are expressed to illustrate the validity of the observed results by the suggested method.
There are several figueres that support to calculations.
The English of the paper is also very good.
In this case, to be honest, I have to recommend it.
Author Response
We are very grateful to the referees for their careful and speedy reading of the manuscripts.
Reviewer 2 Report
My recomendation: the proofs of the therems to shift in Appendix
Reviewer 3 Report
The paper introduces a new algorithm to obtain an approximate solution for a canonical correlation analysis problem.
The manuscript has serious flaws. First, the English must be improved. In fact, sometimes it is quite difficult to understand some sentences or paragraphs. Second, it is quite hard to understand the text. That is, the authors use a confusing and misleading language. Furthermore, some mathematical steps are not trivial and quite confusing for any reader. For instance, why equation (10) implies the existence of the matrices $D_A$ and $D_B$? It would be very helpful if authors provide suitable references (or references to the suitable theorems) when the mathematical reasoning requires so. Third, the mathematical expressions must also be checked. For instance, equation (7) in Theorem 1 is not written right.
Other issues are listed below:
*Some mathematical expressions are not well-written. For instance, page 1 introduction, “S_a \leftarrow S_a - \frac{1}{d} S_a \mathbf{1}_d “ must be “ S_a \leftarrow S_a - \frac{1}{d} (S_a \mathbf{1}_d ) \mathbb{I}_d ”, where $\mathbb{I}_d$ is the dxd identity matrix.
*Introduction (in general): I miss some context of this work. The introduction is not clear enough about the mathematical context of this paper, neither points out the relevance or novelty of the new methods that the authors exhibit.
*In page 1, “In most cases, only one pair of canonical weight vectors is not enough.” Why? In which cases? Can the authors provide some relevant references that support this fact?
*In page 2, lines 30-34: what is the idea behind the Jacobi-Davidson methods? Any non-expert reader may feel uncomfortable about the importance of these methods, specially how and where to apply them.
*Page 2, equation (3): is it necessary to introduce the matrices K and M, and the vector z?
*Check expressions, for instance, in page 2, “canonical correlation generalized eigenvalue (CCGEP)” must read “Canonical Correlation Generalized Eigenvalue (CCGEP)”.
*Preliminaries (in general): these preliminaries are quite confusing, in general. In my opinion, any preliminaries must be self-contained; in the sense that this section must be read and understood without the necessity of reading any other part of the work. Please, make the preliminaries more accurate.
*Following the previous point, in some parts of the paper appears some mathematical elements that have been previously defined but no reference to them is made. Hence, for any reader, the text becomes a puzzle. For instance, in the proof of Theorem 1, it appears: “ U = [x_1, … ] “, but the reader must guess that the “x_i” were the eigenvectors of $\tilde{C}$.
*Theorem 1: it must be rewritten. It is quite hard where the maximization takes place -among all the matrices A and B, or among all the matrices U and V-. Besides, the objective function is not clear; please, check equation (7). The proof must also be rewritten. Please, make the mathematical arguments understandable. It is quite hard for any reader to follow the steps in the proof (and also throughout the text).
*Section 3.1. Strictly speaking equation -not equality- (10) must read “image(C |_Y) \subseteq image (A|_X)” (although the authors may use the typical formula “by an abuse of notation”).
*Page 5, line 54: “are usually not available, only approximate ones”. In which sense are approximates? Why does this fact occur in practice? Can the authors provide some reference?
*Equation (12) is unintelligible. Suddenly, $span(\tilde{U}_j)$ appears and it has not been defined previously. Please, any new mathematical expression must be defined (before or immediately after it appears).
*Theorem 2: I’d rather prefer that this theorem was self-contained, but the mathematical equations. “Adopt the notations so far in this section” is not an accurate expression for any theorem. Furthermore, the notation is quite confusing and hidden throughout the text. Any reader must seek deeply in the text to find the “notation” that Theorem 2 refers to.
*When an equation defines a new mathematical object, please, make use of “ := “ instead of “ = “ (for instance, the expression between the equations (23) and (24)).
I think that the authors need to polish the manuscript quite deeply; this would improve substantially the paper, making it more near to a publishing stage.

Author Response
Please see the attachment。

Round 2
Reviewer 3 Report
The authors have changed the manuscript substantially. Now, it has become more readable. Some minor corrections of English and sentences. For instance, the proof of Theorem 1. Here, the first sentence seems a bit confusing.